# Effect of Protein Supplementation on Orthostatic Hypotension in Older Adult Patients with Heart Failure

**DOI:** 10.3390/geriatrics10020042

**Published:** 2025-03-13

**Authors:** Gohar Azhar, Amanda K. Pangle, Karen Coker, Shakshi Sharma, Jeanne Y. Wei

**Affiliations:** Department of Geriatrics, Donald W. Reynolds Institute on Aging, University of Arkansas for Medical Sciences, Little Rock, AR 72205, USA; amandakpangle@uams.edu (A.K.P.); kcoker@uams.edu (K.C.); ssharma@uams.edu (S.S.); weijeanne@uams.edu (J.Y.W.)

**Keywords:** cardiovascular health, dietary supplements, protein supplements, aging

## Abstract

**Purpose:** Heart failure (HF) impairs physical performance and increases the incidence of orthostatic hypotension (OH). Individuals with OH have a higher risk of falls, which are a major source of morbidity and mortality in older adults. Dietary protein supplementation can improve physical performance in healthy older adult individuals; however, its effect on OH in older adult patients with HF is unknown. **Methods:** Twenty-one older adult patients with mild-to-moderate HF were randomized to placebo or protein supplementation. Dietary protein was supplemented with whey protein so the total protein intake for each participant was 1.2 g/kg bodyweight/day, plus 1 g/day of the amino acid l-carnitine for 16 weeks. Susceptibility to OH was assessed using a head-up tilt test, blood markers, and a functional test (6 min walk) at baseline and 16 weeks. **Results:** There were no differences in tilt test responses or 6 min walk test (6MWT) distances. The protein-supplement group had a significant increase in 6MWT pulse pressures post-walk after 16 weeks of treatment as compared to placebo. However, the tachycardia observed at baseline after 6MWT in the protein group was not seen at the end of the study. There was also a trend towards lower levels of brain naturetic peptide (proBNP) in the protein group vs. placebo at 16 weeks. **Conclusions:** The improved pulse-pressure response to exertion and positive trends in proBNP in this pilot study suggest that dietary supplementation may improve cardiovascular function and general health in individuals with HF and that larger future studies are justifiable.

## 1. Introduction

Orthostatic hypotension (OH) is associated with falls in older adult populations [1,2,3,4,5,6,7,8,9]. Falls often result in loss of health and functional independence and represent a significant economic loss and growing burden on the health care system. In the United States, one quarter of people 65 years and older fall each year, resulting in 3 million emergency room visits, over 800,000 hospitalizations and 300,000 hip fractures, which lead to significant medical care costs for both the patients as well as their families [10]. Older adult patients with heart failure (HF) are particularly susceptible to OH, with a prevalence as high as 83% in hospitalized patients [11].

OH is a concern for patients with HF as they have impaired autonomic responses to stress [12,13]. A reduced ability to increase blood flow or heart rate (HR) appropriately in response to demand can limit cardiac and functional capacity, leading to OH. This inadequate cardiovascular response can be further exacerbated due to poor muscle tone in lower extremities, sarcopenia, and compromised energy utilization in the heart [14,15,16]. These factors are, in part, affected by the nutritional status [16,17].

Protein malnutrition is common in chronic conditions such as HF, as well as in older persons [9,18]. Previous randomized control studies have shown that increasing dietary protein has a clear benefit on lean mass gain and in leg strength, especially when combined with resistance exercise [9,18]. In a recent study of older adult individuals with low physical functioning who were either supplemented with whey protein, targeted essential amino acids or received no supplementation, 6 min walk distances showed an improvement in both supplement groups, while the results for the control (no supplementation) group worsened [9]. The essential amino acid group also showed improvements in grip strength and leg strength, and reduced body weight and fat mass. Importantly, the gains were made without exercise, as older adult HF patients may not be able to maintain the exercise level and gains from exercise in older adults are often lost when exercise is stopped [19].

While protein supplementation helps improve skeletal muscle mass, l-carnitine aids in fatty acid oxidation and energy production in mitochondria during exercise, and has been shown to improve fatty acid metabolism in both a mouse model of aging, as well as in older adult patients with HF [16,20]. Furthermore, l-carnitine supplementation has been shown to reduce systemic inflammation, as reflected by decreases in c-reactive protein, interleukin-1β, and interleukin-6 [21]. These studies suggest that nutritional therapy may be an effective approach to improve the general health of HF patients in a way that is sustainable for individuals with low physical functioning and reduced ability to initiate or maintain exercise.

The goal of the present study was to test the hypothesis that 16 weeks of dietary supplementation with high-quality whey protein and the amino acid l-carnitine would improve physical function and reduce the susceptibility of older adult patients with HF to OH. Tilt response was examined due to the relationship between OH and falls and the high prevalence of OH in older adult patients with HF [1,2,3,4,5,6,7,8,9,11]. The primary endpoints included physical performance, as well as blood pressure (BP) and HR responses to positional change; biochemical measures of general cardiovascular health and HF were also evaluated as secondary measures.

## 2. Materials and Methods

### 2.1. Subjects

A total of 57 potential subjects, aged 60–90 years old, were screened from the geriatric outpatient primary care clinic at the University of Arkansas for Medical Sciences (IRB# 202617). The pre-screen assessment included a brief medical history review for HF status and age confirmation. Inclusion criteria included all races/ethnicities, of any gender, with mild to moderate HF (New York Heart Association class, NYHA 1, 2, or 3).

Exclusion criteria included the following: mild–moderate dementia (as determined by a Montreal Cognitive Assessment, MoCA score < 20); active inflammatory bowel disease; active cancer or chemotherapy within the past year; documented insulin-dependent diabetes mellitus or uncontrolled diabetes with hemoglobin A1C > 8.5%; metoprolol (>50 mg), daily atenolol or more than 12.5 mg twice daily carvedilol; use of psychiatric medications; history of renal failure (stage IV or V); severe orthostasis (>20 mm Hg systolic drop); hip/knee prostheses; carotid artery stenosis; high-grade second-degree heart block; history of arrhythmias (with the exception of stable and controlled atrial fibrillation), history of strokes, Parkinson disease and venous insufficiency; currently in hospice; currently receiving nutrition via feeding tube; and self-reported allergy to whey protein.

Subjects were enrolled if all the inclusion criteria were met and none of the exclusion criteria were identified. The University of Arkansas for Medical Sciences Institutional Review Board reviewed and approved the study and consent forms prior to enrollment (IRB# 202617, 17 January 2023). The study conformed to the Declaration of Helsinki and was retrospectively registered at Clinicaltrials.gov with registration number NCT06256276, 2 December 2024, retrospectively registered.

### 2.2. Experimental Design

#### 2.2.1. Overview

The pilot study was a randomized, placebo-controlled study conducted over 16 weeks (Figure 1), with a homogeneous study population of all subjects with HF (NYHA 1–3). A random number generator was used to randomize subjects to placebo vs. supplement treatment. Subjects were assessed for hemodynamic changes in BP and HR in response to head-up tilt test, biomarkers of cardiovascular function, and physical function at baseline and after 16 weeks of whey protein and l-carnitine supplementation or placebo.

#### 2.2.2. Dietary Supplementation

Protein Supplementation: The intervention group received commercially available whey protein powder (Dymatize Elite Whey, Europa Sports Products, Charlotte, NC, USA) and l-carnitine capsules (Dymatize^®^, Europa Sports Products). The daily amount of protein powder depended on the weight of the patient and protein consumption obtained from food. Each participant completed nutritional assessments and kept a 3-day daily diary of their food intake. The study physician then reviewed the assessments and 3-day daily diary with the subjects to evaluate their nutritional intake and understanding prior to receiving their assigned study supplement. Total daily protein (TDP) requirements for every subject were determined by multiplying the subject’s body weight in kilograms by 1.2 g. The actual protein intake (API) of the subject was calculated by reviewing their 3-day dietary intake. The daily protein deficit was calculated using TDP-API in grams. If the protein deficit was calculated to be 30 g or less, one scoop of protein powder per day was assigned as a supplement. If the protein deficit was calculated to be 31–60 g, then 2 scoops of protein powder per day were assigned as a supplement. The study physician then reviewed the assessments and 3-day daily diary with the subjects. No subject had a daily protein deficit of more than 60 g. Participants were also required to take two l-carnitine capsules (1 g/day) capsules daily. One scoop of protein powder delivered 25 g of protein.

Placebo control: The placebo group received placebo products similar in size, shape, and dosage form to the nutritional supplements. Participants received maltodextrin (Thick-it^®^, Medline, Northfield, IL, USA) in place of whey protein and psyllium husk caps (Wal-Mucil^®^, Walgreens, Deerfield, IL, USA) as a placebo for l-carnitine capsules.

Study supplement was dispensed during the first study visit (study day 0) after randomization and then as needed at subsequent study visits for participants to maintain compliance. Participants dissolved whey or placebo powders in their choice of beverage and were provided with diet logbooks to maintain a daily log of all meals. Logs were collected once per month at each study visit for the following 4 months and were reviewed by both study staff and a research dietician to ensure subject compliance and to keep track of the participants’ daily caloric and protein intake.

#### 2.2.3. Hemodynamic Changes

Autonomic responses and susceptibility to OH to changes in position were assessed by head-up tilt table test before and after 16 weeks of daily supplementation or placebo (Figure 2). Participants were positioned supine on a motorized tilt table for 10 min, then inclined to 60° over 30 s and maintained upright for a period of 20 min before being returned to the supine position for 15 min (Figure 2). HR and BP were measured at 0 and 10 min while the participant was supine, at 15, 20, 25, and 30 min while at 60° tilt, then again while supine at 38 and 45 min (Figure 2, arrows). Blood was drawn for biochemical analysis immediately before and at the end of tilt (Figure 2, asterisks). EKG was continuously monitored throughout the tilt procedure. BP and HR were captured at three points during the tilt procedure (pre-tilt (10 min supine point), during tilt (30 min tilt point), and during recovery (45 min supine point)) and used for analysis while measures at other intervals were used to monitor the condition of the participants during orthostatic stress.

#### 2.2.4. Biochemical Assessment

Blood (10 mL) was collected pre- and post-treatment during the tilt table procedure at 10 min of supine rest immediately before tilt and after tilt at the 30 min time point (Figure 2). Blood samples were evaluated for levels of vasopressin (ADH) and pro B-type natriuretic peptide (proBNP) by LabCorp, Inc. (Little Rock, AR, USA). As this was a pilot study, subjects were not evaluated for blood glucose or insulin levels; all blood samples were collected via random sampling.

#### 2.2.5. Functional Assessment

A 6 min walk test was used to assess changes in physical ability and tolerance to exertion pre- and post-treatment. Participants were instructed to walk as quickly as possible along a straight corridor for 6 min. The total distance traveled, as well as BP and HR before and after the 6 min walk, were recorded.

#### 2.2.6. Body Composition

Body composition was measured using dual-energy X-ray absorptiometry (DXA) to assess changes in lean body mass and fat mass pre- and post-treatment.

### 2.3. Statistical Analyses

Data are expressed as mean (±SE). Comparisons between treatment groups as well as pre- and post-study measures were compared using a Mann–Whitney U Test with α = 0.05 significance level applied to a one-sided hypothesis. In box and whisker plots, boxes represent the inner quartile range and whiskers represent minimum and maximum values.

The mean and standard error for systolic blood pressure (SBP), diastolic blood pressure (DBP), and HR measurements collected during the tilt table procedure were calculated for each intervention group and measurement time point, categorized by baseline and post-treatment (final) visits. Generalized linear mixed models were used to test fixed effects and to estimate intervention group differences for each hemodynamic parameter change from baseline to the post-treatment visit per corresponding time point, with eight measurements per visit. The models included fixed effect terms for the intervention group, measurement time point, and the group–time interaction with participants as the random effect. Placebo and the change from baseline, represented by the study visit’s first measurement, were the reference levels. Model estimates for least squares means were produced for the comparison of groups for each measurement time. The significance level was set at 0.05. The choices of distribution and link for each model were initially narrowed down by examining histograms with distribution overlays and residual Q-Q plots, primarily, and normality tests. Covariance structure was determined empirically and through preliminary model comparisons, although unstructured covariance was expected given the nature of the study. The final determination of models was assisted through examining the residuals and corrected model fit criteria. The analysis was performed using SAS 9.4.

## 3. Results

### 3.1. Demographics

Twenty-one participants consented and were randomized to a treatment group (Figure 1). Twenty participants successfully completed the 16-week study period. One participant in the placebo group was unable to complete the study and was not included in the final analysis. The participants’ demographics and health conditions (comorbidities/medications) are summarized in Table 1 and Table 2, respectively.

### 3.2. Hemodynamic Response to Head-Up Tilt Test

A standard head-up tilt table test was employed to assess hemodynamic responses (Figure 2). SBP dropped during tilt in the placebo group both pre- and post-treatment but remained flat in the supplement group (Figure 3). There were no significant differences in SBP between pre-tilt, tilt, and post-tilt points between baseline and post- treatment within or between treatment groups (Table 3, Figure 3). Changes in DBP to tilt were more varied. In the placebo group, DBP decreased in response to tilt at baseline but increased post-treatment, while the protein supplement group showed the opposite results. However, the only significant difference was an increase in DBP between pre-tilt and tilt points for the placebo group post-treatment (placebo post-treatment pre-tilt: 79 ± 3 vs. placebo post-treatment tilt: 84 ± 2 mmHg) (Table 3).

PP dropped in both groups during tilt both pre- and post-treatment; however, there were no significant differences between pre-tilt, tilt, and recovery points within or between treatment groups (Table 3). HR increased in both groups during the tilt point both pre- and post-treatment (Figure 3). HR at baseline was significantly higher post-treatment compared to at baseline in the placebo group (placebo baseline pre-tilt: 62 ± 2 vs. placebo post-treatment pre-tilt: 70 ± 3 mmHg) and between the placebo and protein groups at baseline (placebo baseline pre-tilt: 62 ± 2 vs. protein baseline pre-tilt: 71 ± 4 mmHg) (Table 3, Figure 3).

### 3.3. Biochemical Assessments

Blood levels of ADH (antidiuretic hormone) and proBNP (propeptide of brain natriuretic peptide) were measured pre- and post-treatment during the tilt table test. Blood was drawn immediately before tilt and after 20 min of tilt (Figure 4). There were no significant differences between groups for either measure (Figure 4A,B); however, there was a trend toward a significance decrease in proBNP level in the protein group between pre- and post-treatment (*p* = 0.08) while levels increased in the placebo group (Figure 4B). ProBNP was also higher post-tilt vs. pre-tilt for both groups throughout the study (Figure 4B).

### 3.4. 6 Min Walk Test

Physical stamina and autonomic function were assessed by a 6 min walk test at baseline (pre-treatment) and post-treatment (final) visits for each group. There were no significant differences in distance walked at baseline compared to post-treatment visits or between treatment groups. There were no significant differences in SBP post-walk to pre-walk; however, there was a trend toward a significant increase in SBP pre-walk to post-walk on the post-treatment visit in the protein group compared to the placebo group (Figure 5A). There were no significant changes in pre- and post-walk DBP between baseline and post-treatment visits or between treatment groups, but the placebo group had a significant increase in DBP between pre- and post-walk at baseline (placebo baseline pre-walk of 74 ± 3.4 vs. placebo baseline post-walk of 85.3 ± 3.5 mmHg) (Figure 5B).

There was no significant difference in pre- vs. post-walk PP in the placebo group at either the baseline or post-treatment visits, but there was a trend toward an increase in PP between the baseline and post-treatment visits. The protein group had a significantly higher PP post-walk compared to pre-walk at the post-treatment visit (protein post-treatment pre-walk of 52.4 ± 2.1 vs. protein post-treatment post-walk of 63.0 ± 3.3 mmHg) as well as compared to the placebo group post-walk at the post-treatment visit (placebo post-treatment post-walk: 53.6 ± 4.0 mmHg) (Figure 5C). HR also showed an increasing trend in the placebo group between baseline and post-treatment visits but the results did not reach significance (Figure 5D). The protein group had a significantly higher post-walk HR at baseline, but after 16 weeks of treatment the post-walk HR in the protein group was not increased.

## 4. Discussion

Orthostatic hypotension (OH) is a common condition in older adults and has a complex etiology. These etiologies can include neurocardiogenic system dysfunction resulting from loss of baroreceptor sensitivity with age, reduced sinus node function, atrial–nodal conduction blocks and autonomic failures from multiple diseases (e.g., diabetes mellitus, Parkinson’s disease), and non-neurogenic causes related to aging, including decreased lower-extremity muscle tone, lower vascular tone, venous insufficiency, sarcopenia, and reduced cardiac output [15,21,22,23]. In addition to these factors, HF patients, who have a high prevalence of OH, often suffer from autonomic dysfunction [11,24]. The primary risk associated with OH in older adults is falling [1,2,3,4,5,6,7,8,9,10]. Injuries suffered during a fall, such as hip fractures and head injuries, are associated with a high risk of mortality [1,2,3,4,5,6,7,8,9,10,11,25,26].

Cardiovascular function is tied to nutrition, and protein malnutrition is common in aging [9,18,27]. Furthermore, patients with chronic medical conditions such as HF are at increased risk of malnutrition, which is associated with poor prognosis [9,28]. The HF pathology itself contributes to malnutrition via several different mechanisms, such as altered gut absorption, hyper catabolism, anorexia, and intestinal inflammation [9]. Various studies have tried to establish the prevalence of protein-energy malnutrition in HF patients, but the exact prevalence remains unknown, with estimates ranging from 8% to 54% [28]. However, studies suggest that dietary macronutrients, specifically dietary protein intake, may affect HF symptoms and outcomes [9,28].

In this current study, we examined the effect of whey protein and l-carnitine supplementation on autonomic response, cardiovascular health, and physical function in older adult HF patients. The principal finding of this study was that daily supplementation with whey protein and l-carnitine for 16 weeks significantly improved PP in response to physical challenge comprising a 6 min walk. Additionally, there was a tendency towards a reduction in proBNP levels, which suggests a reduced risk of heart failure [29,30].

Whey protein was chosen as the protein source because it is the most commonly used and effective protein supplement to support skeletal muscle synthesis and is widely available to consumers [31]. The daily protein goal of 1.2 g/kg bodyweight/day was based on the optimal dietary protein requirements to improve muscle health and avoid sarcopenia in older adults [32]. In the older adult population, 1.0–1.3 g/kg bodyweight/day appears to be the required dietary protein intake to maintain optimal physical function. Additionally, there is ample evidence of whey supplementation improving skeletal muscle mass in older adults [9,28,31,32,33]. However, there is less evidence that whey protein alone can improve cardiac muscle function [9]. Many studies have shown that abnormalities in skeletal muscle due to HF can further limit functional capacity in HF [9,28,31,32,33].

l-carnitine was supplemented to improve metabolism and muscle health [16,20]. Our dose of 1.0 g/day is consistent with studies showing an improvement in skeletal muscle and general health for a variety of patients and conditions [9,28,31,32,33,34]. In older adults, 1.5 g/day reduced frailty indicators, while 1.5 to 2 g/day reduced fatigue, improved physical performance, and increased muscle mass [35,36,37]. While the higher dose of 1.5–2 g/day in certain studies led to more improvements than the smaller 1 g/day dose, we chose to go with the minimal effective dose of 1 g/day to better reflect actual clinical practices in the older adult population, where treatments are slowly introduced according to tolerability. Studies have shown that l-carnitine supplementation is also beneficial in HF. A meta-analysis of 17 randomized controlled trials of l-carnitine treatment in chronic HF showed improvements in cardiac output and proBNP levels at doses of 1.5 to 6 g/day [38].

Patients with HF often have impaired autonomic responses, which can lead to OH [9,13,39]. The effect of supplementation on autonomic function was assessed by a head-up tilt table testing. Neither group in our study met the definition for OH (sustained decrease in SBP ≥20 mmHg or ≥120% of baseline and/or DBP ≥10 mmHg when in upright position) [40]. We did not include subjects with a history of OH in our study as we wanted to minimize the incidence of syncope during tilt. However, a degree of underlying autonomic dysfunction may be suggested in both groups because both had concurrent elevated HR with a drop in SBP in response to tilt, which is more characteristic of autonomic dysfunction than a vasovagal reflex response [1,2,3,4,5,6,7,8,9,41]. Interestingly, the peak drops in SBP and increases in HR occurred at 20 min rather than within the first 3 min of the tilt test, suggesting susceptibility to delayed OH rather than the classical (initial onset) OH typically seen in older patients [40,41]. A study by Oberoi et al. [42] observed that, for older adults, consuming 30 g and 70 g of whey protein drink exhibits a beneficial effect on SBP between 120 and 180 min after ingestion.

The levels of circulating ADH and proBNP were measured at baseline and after 16 weeks as indicators of cardiovascular health related to OH and HF. Additionally, classical OH is associated with increased ADH during tilt, and proBNP is a marker of cardiomyocyte damage [40,43]. While strong trends were observed, neither marker reached statistical significance. Both groups had approximately 50% lower levels of ADH at the end of the study. While this change suggests either reduced potential autonomic dysfunction or worsening HF, it cannot be attributed to the treatment. ProBNP levels increased by ~65% in the placebo group but decreased by ~37% in the supplement group. Because proBNP is used as an indicator of damage to cardiomyocytes, it is reasonable to regard the decrease as a possible improvement in the disease process, which is also supported by the fact that the placebo group continued to worsen despite the non-significant results [43].

The 6 min walk test was used to evaluate the effect of supplementation on the ability of older adult HF patients to perform basic daily physical activities. This standardized test has been used as a practical and convenient assessment of cardiopulmonary fitness and physical function in adults [9]. While the distance traveled did not improve with supplement therapy, there was a significant augmentation in post-walk PP at the end of the 16 weeks in the supplement group. This is supported by our previous findings, where we observed an improvement in 6MWT in older adults consuming EAA or whey protein for 12 weeks [44]. PP augmentation in response to exercise is associated with increases in cardiac output and vascular compliance and is inversely associated with HF and all-cause mortality [45]. A poor cardiac output and reduced vascular compliance are contributing factors for increased risk of OH in aging [45,46]. Additionally, in the protein group, post-walk tachycardia was observed at baseline, which was not present post 6MWT at the end of the study. This suggests better physical performance after 16 weeks in the protein group.

## 5. Limitations

This study was a single-center, pilot study with a small sample size. Furthermore, this study had a short intervention time of 16 weeks, and a longer-term intervention and follow-up of at least 6 months would have been useful. This study included only whey protein and l-carnitine supplementation’s effects on OH; these observations cannot be applied to other protein sources.

## 6. Conclusions

The results of this pilot study provide preliminary evidence that protein and targeted amino acid supplementation in older adult HF patients may improve measures of cardiovascular health and physical function, which are contributing factors to OH and fall risk with advancing age. The positive trends across multiple measures, combined with the simplicity of the intervention and its compatibility with other therapies, suggest that additional investigation of this form of dietary intervention in larger studies might be impactful.

## 7. Future Directions

Older patients are often nutritionally deficient in protein intake for various reasons, such as difficulty chewing, digestive issues, and the higher cost of protein-rich food. This reduction in the amount of protein in the diet not only impacts muscle mass but might also influence the autonomic control of blood pressure and heart rate. Larger, randomized controlled trials are needed to assess the value of protein supplementation on overall cardiovascular health and the strength and functioning of older adults.

## Figures and Tables

**Figure 1 geriatrics-10-00042-f001:**
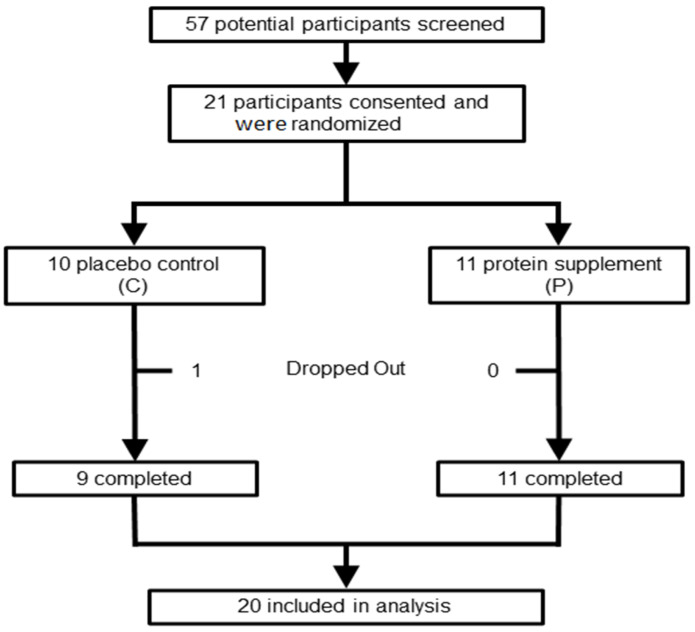
Schematic representation of study.

**Figure 2 geriatrics-10-00042-f002:**
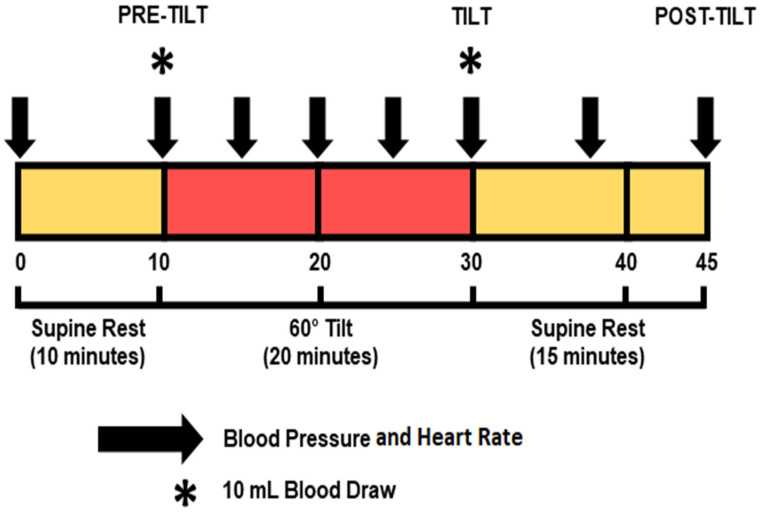
Tilt-table procedure. Blood pressure and heart rate were obtained at 0 and 10 min while supine, every 5 min at 60° tilt from 10 to 30 min, and while supine at 38 and 45 min (arrows). Baseline, tilt and recovery points were used for analysis. Blood was drawn immediately before tilt and after tilt (asterisks) for vasopressin and pro-BNP levels. The yellow color indicates the supine rest time (10, 15 min) pre and post tilt, and red color indicate the tilt time (20 min).

**Figure 3 geriatrics-10-00042-f003:**
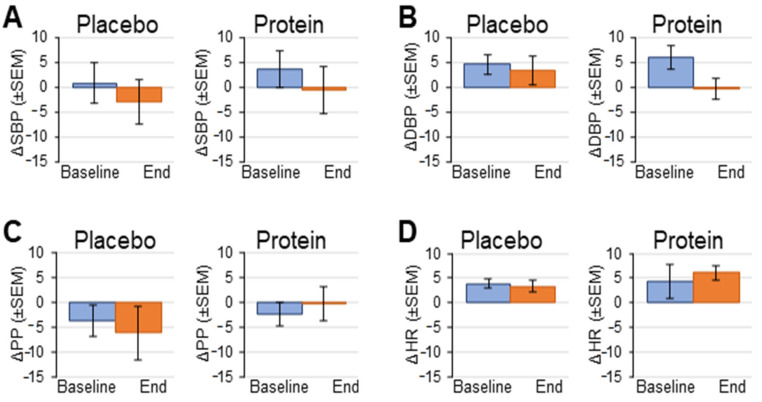
Group hemodynamic changes in tilt (from pre-tilt to tilt) at study baseline and at the end of 16 weeks of protein supplementation or placebo control. (**A**) Systolic blood pressure change (SBP); (**B**) diastolic blood pressure change (DBP); (**C**) pulse pressure change (PP); (**D**) heart rate change (HR). Data are means ± SE.

**Figure 4 geriatrics-10-00042-f004:**
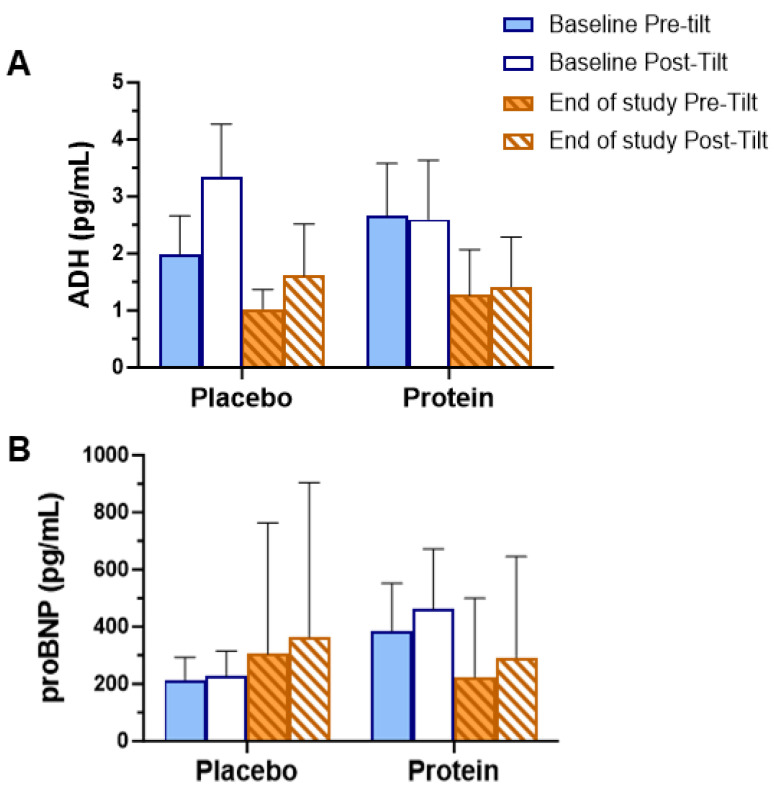
Cardiovascular health blood markers before and after 16 weeks of protein supplementation or placebo control. (**A**) Anti-diuretic hormone, ADH; (**B**) pro B-type natriuretic peptide, proBNP.

**Figure 5 geriatrics-10-00042-f005:**
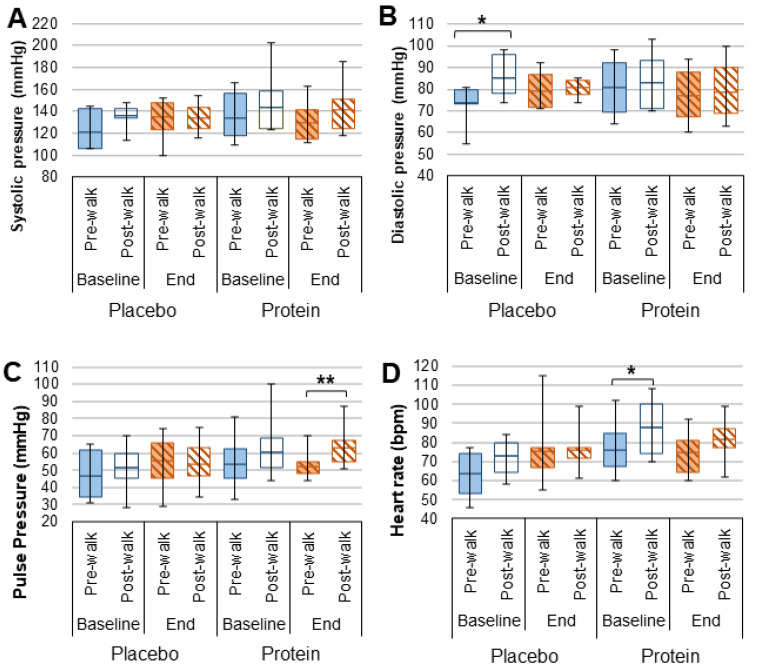
(**A**–**D**) Hemodynamic changes before and after 6 min walk tests performed before (baseline) and after (final) 16 weeks of protein supplementation or placebo. Significant difference from pre-walk: * *p* < 0.05; ** *p* < 0.01. There were no differences in distances walked.

**Table 1 geriatrics-10-00042-t001:** Treatment group demographics.

	Control	Protein	All
Total Participants	9	11	20
Age			
Mean (SE)	75 (2.5)	74 (1.7)	75 (1.4)
Median	75	74	75
Mode	75	79	74
Range	64–86	63–81	63–86
Gender			
Male	1	2	3
Female	8	9	17
Race			
Asian	0	1	1
African American	1	1	2
White	8	9	17

**Table 2 geriatrics-10-00042-t002:** Participant characteristics.

Category	Specific Factor	Control (*n* = 9)	Protein (*n* = 11)
	CAD/Stents	0	2
	HTN	7	10
	HLP	6	8
	AFib (stable and controlled)	0	1
	DM	0	2
	Pacemaker/defibrillator	1	1
	COPD/lung diseases	0	1
Medications	ACEI/ARB	6	5
	Beta Blockers	4	2
	CCB	4	3
	Diuretic	2	2
	Anti-arrhythmic	0	0
	Nitrates	1	0
	Statins	5	4
Other	Fall	1	2
	Dizziness	1	3

ACEI = angiotensin-converting enzyme inhibitor; AFib = atrial fibrillation; ARB = angiotensin II receptor blocker; CAD = coronary artery disease; CCB = calcium channel blockers; COPD = chronic obstructive pulmonary disease; DM = diabetes mellitus; HLP = hyperlipidemia; HTN = hypertension.

**Table 3 geriatrics-10-00042-t003:** Tilt table test hemodynamics.

	Baseline			End of Study		
	Pre-Tilt	Tilt	Post-Tilt	Pre-Tilt	Tilt	Post-Tilt
	Placebo Control					
SBP (mmHg)	135 ± 5	130 ± 8	137 ± 5	132 ± 6	125 ± 7	136 ± 6
DBP (mmHg)	80 ± 2	80 ± 4	87 ± 4	79 ± 3	84 ± 2 ^†^	84 ± 4
PP (mmHg)	54 ± 4	50 ± 5	50 ± 6	53 ± 4	47 ± 5	52 ± 3
HR (beats/min)	62 ± 2	69 ± 3	63 ± 3	70 ± 3 *	77 ± 6	68 ± 4
	Protein Supplement					
SBP (mmHg)	132 ± 4	132 ± 5	132 ± 6	136 ± 6	132 ± 7	132 ± 7
DBP (mmHg)	77 ± 4	80 ± 4	79 ± 5	83 ± 4	82 ± 4	81 ± 4
PP (mmHg)	56 ± 2	52 ± 3	52 ± 3	53 ± 3	50 ± 3	50 ± 4
HR (beats/min)	71 ± 4 ^‡^	75 ± 5	64 ± 4	72 ± 4	78 ± 3	70 ± 3

Group hemodynamic changes to tilt before and after 16 weeks of protein supplementation or placebo control: systolic blood pressure (SBP); diastolic blood pressure (SBP); pulse pressure (PP); and heart rate (HR). Data are means ± SE. * Difference from baseline; ^†^ difference from pre-tilt; ^‡^ difference between treatment groups: *p* < 0.05.

## Data Availability

The raw data are available without reservation upon reasonable request.

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
