# Peer review of "Effect of Protein Supplementation on Orthostatic Hypotension in Older Adult Patients with Heart Failure"

_geriatrics, 2025, doi:10.3390/geriatrics10020042_

Round 1

Reviewer 1 Report

Comments and Suggestions for Authors

The authors presented a randomized controlled trial about the possible benefit of whey protein and L-Carnitine supplementation in older adults with heart failure. Evidences showed that whey protein has been shown to increase muscle mass and microvascular function, whilst L-carnitine may increase heart function and functional capacity. 
As indicated by Julious, the minimum sample size in a pilot randomized controlled study should be 12 vs. 12 (10.1002/pst.185); please elucidate the choice of your sample size. Moreover, authors should consider a possible paragraph of limitations: duration and possible side effects (10.3390/nu13103453) of nutritional implementation in older adults.

Author Response

We thank reviewer #1 for his/her thorough review of the manuscript and are deeply grateful for their thoughtful comments and suggestions. Our responses are enumerated below, and we have also addressed/highlighted the comments in the text of the manuscript where applicable.

  1. The authors presented a randomized controlled trial about the possible benefit of whey protein and L-Carnitine supplementation in older adults with heart failure. Evidences showed that whey protein has been shown to increase muscle mass and microvascular function, whilst L-carnitine may increase heart function and functional capacity.

As indicated by Julious, the minimum sample size in a pilot randomized controlled study should be 12 vs. 12 (10.1002/pst.185); please elucidate the choice of your sample size. Moreover, authors should consider a possible paragraph of limitations: duration and possible side effects (10.3390/nu13103453) of nutritional implementation in older adults.

We agree with the author about the small sample size in our study. Initially in this study we screened 57 potential subjects, however not all participants met the inclusion and exclusion criteria. As recommended by the reviewer, we have added a paragraph of limitations associated with our study.

Reviewer 2 Report

Comments and Suggestions for Authors

I congratulate the authors for their work!

Introduction is comprehensive, provides enough information about the topic. The aim of the study is well described in Introduction section.

Here some suggestions:

Title: please change the title - the aim of study was the effect of protein suplimentation on ortostatic hypotension in patients with heart failure.

Abstract : please change the purpose: again the aim of study was the effect of protein suplimentation on ortostatic hypotension in patients with heart failure.

Methods and materials: including/exclused criteria : were patients with arthrosis, hip/knee prostheses included? carotid artery stenosis has been ruled out? psychiatric medication (with adverse effects of orthostatic hypotension? Stroke, Parkinson disease? Venous insufficiency?

- The history of arriythmias was mentioned at exclusion criteria and it is a patient with atrial fibrillation included.

-please include the demographic data, participant characteristics in the Result section, not in Methods and materials section

Results: -please remove ” Blood  was drawn immediately before tilt and after 20 minutes of tilt” (line 231) from Results section – the blood sample collection was mentioned in Material and methods section

Discussion:

-please add some comments and comparation with results of other studies

-add limitations of the study

-add future directions

Author Response

Reviewer #2

We thank reviewer #2 for his/her thorough review of the manuscript and are deeply grateful for their thoughtful comments and suggestions. Our responses are enumerated below, and we have addressed/highlighted the comments in the text of the manuscript where applicable.

  1. I congratulate the authors for their work!

Introduction is comprehensive, provides enough information about the topic. The aim of the study is well described in Introduction section.

We Thank reviewer for the positive feedback for our study.

  1. Title: please change the title - the aim of study was the effect of protein supplementation on orthostatic hypotension in patients with heart failure.

We have revised the title as suggested by the reviewer.

  1. Abstract: please change the purpose: again, the aim of study was the effect of protein supplementation on orthostatic hypotension in patients with heart failure.

We have changed the purpose as suggested.

  1. Methods and materials: including/exclusion criteria: were patients with arthrosis, hip/knee prostheses included? carotid artery stenosis has been ruled out? psychiatric medication (with adverse effects of orthostatic hypotension? Stroke, Parkinson disease? Venous insufficiency?

We excluded patients who had hip/knee replacement, carotid artery stenosis and were on psychiatric medication. We also excluded patients who had stroke, Parkinson and venous insufficiency. We have included these conditions in the exclusion criteria in the Material and Method section (Line 86-90).

  1. The history of arrythmias was mentioned at exclusion criteria and it is a patient with atrial fibrillation included.

We included a patient with atrial fibrillation, however the patient with atrial fibrillation was stable the atrial fib. was very well controlled. We changed and highlighted this in Table 2.

  1. Please include the demographic data, participant characteristics in the Result section, not in Methods and materials section.

We have moved the demographic data and participant characteristics [Table 1 and 2] in the Result section as suggested by the reviewer.

  1. Results: -please remove” Blood was drawn immediately before tilt and after 20 minutes of tilt” (line 231) from Results section – the blood sample collection was mentioned in Material and methods section

We have removed this line from our manuscript.

  1. Discussion: please add some comments and comparation with results of other studies

As suggested, we have added comparison studies in our discussion. One study by Oberoi et al [2022] reference no. 42 [line no. 325-327] and another by Azhar et al [2024] reference no. 44 [line no 344-346] have been added.

  1. Add limitations of the study

We have added a paragraph highlighting the limitations associated with our study [355-359].

  1. Add future directions

As suggested, we have added Future directions in a separate section in our manuscript [372-377].

Round 2

Reviewer 1 Report

Comments and Suggestions for Authors

The authors added a paragraph about limitations.

Reviewer 2 Report

Comments and Suggestions for Authors

The authors responded to all my suggestions. The manuscript is interesting and deserves to be published.